# Decreased Saccadic Eye Movement Speed Correlates with Dynamic Balance in Older Adults

**DOI:** 10.3390/ijerph19137842

**Published:** 2022-06-26

**Authors:** Youngsook Bae

**Affiliations:** Department of Physical Therapy, College of Health Science, Gachon University, 191 Hambangmoe-ro, Yeonsu-gu, Incheon 21936, Korea; baeys@gachon.ac.kr; Tel.: +82-32-820-4324

**Keywords:** saccadic eye movement, dynamic balance, elderly

## Abstract

This study aimed to determine the change in saccadic eye movement (SEM) speed according to age (young older; 65–72 years, middle older; 73–80 years, old older: over 81 years) in the elderly and identify the correlation between SEM speed and balance ability. We recruited 128 elderly individuals and measured their SEM speed and balance. The SEM speed was measured to allow the target to appear once every 2 s (0.5 Hz), twice per second (2 Hz), or thrice per second (3 Hz). The SEM performance time was 1 min with a washout period of 1 min. Balance ability was measured using the functional reach test (FRT), timed up-and-go test (TUG), and walking speed (WS). As age increased, FRT, TUG, and WS decreased and SEM speed was significantly decreased in old older than in young older adults at 3 HZ. In all participants, the 3 Hz SEM speed was significantly correlated with TUG and WS. Therefore, SEM speed may be inadequate or decreased in response to rapid external environmental stimuli and may be a factor that deteriorates the ability to balance in older adults.

## 1. Introduction

The visual system is the most important sensory organ in maintaining balance. In the elderly, visual function is closely associated with the ability to maintain balance and perform daily activities independently [1]. With age, the visual system function deteriorates, and eye movements also decline [2]. Eye movements are induced to assess surroundings and respond to visual stimuli. Voluntary eye movements, such as pursuit eye movement (PEM) and saccadic eye movement (SEM), are used to track moving objects [3], and they can influence posture control in an upright posture [4,5]. In particular, postural sway is reduced during SEM in older people [5,6].

SEM is a rapid eye movement that changes the fixation point of the gaze; its preparation and performance are determined by the coordinated operation of many brain structures, including brainstem structures and different areas of the cerebral cortex [7]. SEM relative to the moving target plays an important role in controlling balance during upright standing body position in space [8,9]. These findings indicate that SEM may be correlated with balance ability.

According to previous studies, SEM characteristics may indicate age-related changes in the nervous system [10,11]. In addition, in healthy adults, the latency of SEM increases significantly after the age of 60 years [12]. In addition, peak saccade velocity was significantly slower for amplitudes exceeding 20 degrees, and saccadic reaction times were prolonged in older adults [13]. An increase in SEM latency implies a decrease in the speed of SEMs in response to visual stimuli from the external environment. This means that SEM is slower in people over the age of 60 years, who may have a decreased balance ability. Despite the abundance of SEM studies, no study evaluated changes in eye movement speed according to the external environmental velocity with age. Additionally, there is no study that has confirmed the correlation between SEM speed and balance ability in the elderly.

Therefore, this study aimed to compare SEM speed according to age in young older (aged 65–72 years), middle older (aged 73–80 years), and old older (aged over 81 years) community-dwelling and independent-living individuals. In addition, we intended to determine the correlation between balance ability and 0.5, 2, and 3 Hz SEM speed in the participants. This study hypothesized that 0.5, 2, and 3 Hz SEM speed decreases with increasing age, and decreased SEM speeds are associated with decreased balance ability.

## 2. Materials and Methods

### 2.1. Design

This study used a cross-sectional, observational design. All participants were given detailed information on the study procedure and safety and provided written informed consent. All study procedures were approved by the institutional review board of Gachon University, and the study was conducted in accordance with the tenets of the Declaration of Helsinki. The data for the study were collected between June 2021 and August 2021.

### 2.2. Participants

The participants were 128 older adults (age range, 65–89 years) who were recruited through telephone interviews and posters at community centers. The participants were selected based on the inclusion and exclusion criteria. Inclusion criteria were: (1) adults aged >65 years with the ability to perform activities of daily living independently, (2) absence of premorbid or current orthopedic problems involving the lower extremities and neurological disorders such as cerebral infarction, and (3) no history of surgery due to musculoskeletal disorders of the lower extremities. Individuals who had a Mini-Mental State Examination score < 24 and those who did not perform the measurement procedures were excluded.

This study used G × Power 3.1.9 software to calculate the sample size, which was determined on the basis of a two-tailed test, power = 0.8, α = 0.05, effect size = 0.5. The calculated sample size was 128 [14].

### 2.3. Materials

#### 2.3.1. SEM Speed

The subjects’ eye movement speed in response to external stimuli was measured using an eye-tracking device. Eye-tracking technology offers the possibility of capturing visual behavior information in real time and obtaining gaze positions within stimuli [15,16]. The eye-tracking device used to measure eye movements was a Tobii X2-30. This eye tracker is a remote system (not head-mounted) that collects 30 gaze data points per second and delivers accurate gaze position data to where the subject is gazing. Data were collected from both eyes, and the accuracy and precision conditions were 0.4° and 0.32°, respectively. The eye tracker can operate at a distance of 70–90 cm from the user. The fixed distance was set at 1 m. The eye tracker unit is 184-mm long and weighs 200 g. It is attached to the bottom of the screen of a 24 inches monitor (LG, South Korea, resolution 1920 × 1080) with a width of 52.8 cm and a height of 29.7 cm for use.

The subject was seated on a chair approximately 80–90 cm in front of the monitor, and the head was fixed using a neck pillow to prevent head movement. Subsequently, the movement of a red dot with a diameter of 2 cm on a white background was tracked by the eye on the monitor. The red dots were generated using Flash software and displayed on the monitor. The red dot appeared in one place on the screen, disappeared, and immediately reappeared in another position, which caused the participants’ eyes to follow the red dot, which appeared randomly in the horizontal and vertical directions. The moving speed of the red dots on the monitor was set to 0.5, 2, and 3 Hz for 50 s, as described in a previous study [9]. The frequency was 0.5 Hz if the target appeared once every 2 s, 2 Hz if it appeared twice per second, and 3 Hz if it appeared thrice per second. The speed was measured for 30 s from 10 s after the start of the eye movement.

A program was prepared to calculate the data obtained from the Tobii tracker unit as eye movement speed. The eye arrival time from dot A to dot B and the distance between the two points were calculated, and the speed was calculated in cm/s. The size of the monitor used in this study was 52.8 cm in width and 29.7 cm in height, and the number of pixels were 1920 × 1080. Based on the screen characteristics, each pixel measured 0.0275 cm. The distance between points A and B was calculated SQRT(POWER((x2 − x1) × 52.8)^2^), POWER((y2 − y1) × 29.7)^2^). The average was then calculated as the sum of the speed values/number of times the dot moved. In this study, the average speeds of the left and right eyes were used.

#### 2.3.2. Balance Ability

Balance ability was measured using the timed up-and-go test (TUG), functional reach test (FRT), and walking speed (WS) to evaluate fall risk.

TUG is a sensitive and specific tool for identifying adults at risk for falls [17]; it evaluates functional mobility in older adults living in the community. The TUG measures the time that a participant needs to seat themselves on an armless chair. In addition, upon the voice signal “start,” they get up from the chair, walk 3 m at a comfortable speed, return to the chair, and sit down in the chair again. A total of three measurements were conducted, and the mean value was calculated and used for the analysis. It has excellent intra-rater reliability for determining the risk of falls in community-dwelling elderly individuals [18].

The FRT is a fall risk screening test [19], in which both feet are on the ground, one arm is extended as far forward as possible, and the distance between the starting and ending positions of the fingertips is measured [20]. In particular, the risk of recurrent falls can be identified if the distance is <25.4 cm in the elderly [21]. The FRT has been developed to evaluate dynamic balance ability related to maintaining postural stability during movement and is closely tied to falls risk in the elderly [22]. It has been reported to have predictive validity for the incidence of falls in the elderly [23]. WS was measured using a 10-m gait test, which is a commonly used measure for assessing WS in older adults [24]. The participants were asked to walk 14 m at their preferred pace. The first and last 2 m distances were set as acceleration/deceleration points, respectively, and were not included in the evaluation. Therefore, we evaluated the WS within the 10 m midsection. The 10-m walk test has demonstrated excellent reliability in many conditions, including in healthy adults, and excellent test–retest reliability for comfortable gait speed (r = 0.75–0.90) [25]. We measured the TUG, FRT, and WS in triplicate and recorded the average measurements.

### 2.4. Procedures

In the beginning of the study, 137 older adults were screened. However, nine participants did not meet the inclusion criteria. Demographic characteristics of the participants, including a history of falls, were examined, and mini-mental state examination (MMSE), balance ability, and SEM speed were measured. The SEM speed was measured at 0.5 Hz, 2 Hz, and 3 Hz. All participants were trained to become familiar with how to perform SEM before measuring SEM speed. Participants performed SEM randomly at 0.5, 2, and 3 Hz, depending on the code shown in a sealed, opaque envelope.

All the data were collected in a university laboratory. Although researchers were aware of the purpose of this study, outcome assessors were blinded to the purpose of this study.

### 2.5. Result Analysis

All statistical analyses were performed using SPSS version 26 (IBM Corp., Armonk, NY, USA). We analyzed the frequency and used descriptive statistics to assess participants’ general characteristics. The measured variables between groups according to age (younger, middle, and old older) were compared using one-way repeated ANOVA. The post-hoc test was conducted using Dunnett. In addition, the correlations between 0.5, 2, and 3 Hz SEM speed and dynamic balance for all participants were analyzed using multivariable linear regression analysis. All variables are expressed as the mean ± SD.

## 3. Results

The average age of the 128 participants (27 men and 101 women) was 72.27 years.

### 3.1. Comparison of SEM Speed and Balance Ability According to Age

As age increased, the TUG (*p* < 0.001) increased and WS (*p* < 0.001) decreased. The FRT was not significant difference. The SEM speed at 0.5 and 2 Hz were not significant, but 3 Hz (*p* = 0.013) more significantly decreased in old older than in young older adults (Table 1).

### 3.2. Multivariable Regression Analyses Predicting Balance Ability with Respect to SEM Speed in Elderly

Table 2 shows the predictors of balance ability among the SEM speeds. Furthermore, 3 Hz was associated with TUG (β = −0.242, *p* = 0.006) and WS (β = −0.282, *p* = 0.001) and was shown to be a predictor. However, there was no significant correlation between the SEM speed and FRT.

## 4. Discussion

In this study, the author compared differences in balance ability and SEM speed with increasing age and identified the correlation between SEM speed and balance ability. As age increased, TUG increased and WS decreased significantly, and the 3 Hz SEM speed decreased significantly in older participants. In addition, 3 Hz SEM speed was confirmed to be a predictor of TUG and WS.

In the upright posture, visual information creates postural synergy and plays an important role in balance control [26]. In particular, eye movements may interact with the balance control system to maintain an upright standing posture in humans [27]. People routinely search their environment using SEMs and/or coordinated eye movements to obtain useful information [28]. As environmental conditions change, signals from vestibular, visual, and somatosensory inputs are reweighted to maintain upright posture [29]. The reweighting process seems to be altered in older adults [30], and even displays stronger coupling between visual information and body sway [30,31].

In this study, the hypothesis was that 0.5, 2, and 3 Hz SEM speeds decreased with age. However, a decrease of 3 Hz was statistically significant in the very old individuals >80 years of age compared with that in older individuals, whereas there was no significant difference between 0.5 Hz and 2 Hz SEM rates. These results suggest that SEM does not respond adequately to fast-moving objects in the environment in very old individuals >80 years of age. The visual system plays a critical role in the control of walking [32], and efficient walking is dependent on the visual information gathered by SEMs [33]. Our findings suggest that TUG significantly increased and WS significantly decreased with age, but SEM speed significantly decreased only at 3 Hz. This means that reduced fast eye movement may be correlated with walking.

Slower WS is associated with age, and WS is a clinical marker and an important measure of functional capacity in the elderly [34]. WS is related to the overall involvement of neural networks in the cognitive domain [35]. This means that if the SEM is slow, WS can also be slowed. In this study, we also identified that there was a correlation between SEM and WS, which was proven in a previous study [33]. Therefore, inappropriate eye movement can negatively affect walking, and there may be a correlation between SEMs and walking. Eye movements are different static states, such as sitting, and dynamic states, such as walking [36]. In the dynamic state, rather than the static state, the speed of the eye movement should change rapidly according to the speed of the person’s movement. In this study, TUG and WS were significantly lower in old older individuals than in young older individuals. Moreover, in multiple regression analysis, β of FRT was 0.083, and TUG and WS were −0.242 and −0.282, respectively, at a 3 Hz SEM speed. These results showed that dynamic balance is a more influential factor for BT at a 3 Hz than at 0.5 or 2 Hz SEM speed. The WS and TUG measure balance ability while moving the base of support (BOS), while the FRT measures balance ability when the BOS is fixed. These results suggest that a decreased speed of fast eye movement may also decrease dynamic balance. In particular, in the elderly, dynamic balance ability may be reduced due to inappropriate or reduced eye movements to rapid SEM stimulation. Therefore, the authors suggest that fast eye movement speed is a more influential factor in dynamic than in static conditions in very old individuals.

This study has several limitations. First, the experiments were not performed in a real environment. Second, the age distribution of the participants was not uniform. Therefore, further research on this topic is required. Despite these limitations, this study has several advantages. This is the first study to identify that SEM speed decreases with increasing age and the correlation between SEM speed and balance ability in the elderly. Therefore, this study provides the basis for further research on SEM interventions to promote or improve balance ability. In addition, this study has clinical significance because balance ability was verified through SEM speed.

## 5. Conclusions

Our study showed that eye movement speed slows and balance ability deteriorates with age. In particular, our findings showed that a 3 Hz speed was decreased in adults aged >80 years, suggesting that they did not respond adequately to fast-moving objects in the environment. Importantly, it was also confirmed that a 3 Hz SEM speed had a significant correlation with TUG and WS. These results suggest that inadequate and reduced SEM by fast-moving objects in the external environment can be a factor that can deteriorate balance ability. We propose that future studies should investigate the relationship between SEM speed and fall risk by confirming the correlation between fall risk and saccade eye movement in older adults with an impaired balance ability and a tendency to fall.

## Figures and Tables

**Table 1 ijerph-19-07842-t001:** Comparisons of variables according to age.

Variables	Young Older (*n* = 43)	Middle Older (*n* = 59)	Old Older (*n* = 26)	Young Older vs. Old Older (*p*-Value)	Middle Older vs. Old Older (*p*-Value)	Between Groups	
						F	*p*
Age (years)	68.95 ± 2.05	76.19 ± 1.99	83.65 ± 3.23			334.769	<0.001
Height (cm)	154.64 ± 7.11	150.88 ± 19.24	154.81 ± 9.07			1.133	0.325
Weight (kg)	61.77 ± 11.50	59.42 ± 8.73	63.59 ± 11.74			1.612	0.204
MMSE (score)	26.16 ± 2.95	26.49 ± 2.31	25.69 ± 3.48			0.746	0.476
**Balance ability**							
FRT (cm)	27.58 ± 6.82	26.31 ± 5.90	24.17 ± 5.91	0.051	0.231	2.432	0.092
TUG (s)	12.19 ± 2.96	12.25 ± 2.33	15.14 ± 3.24	<0.001	<0.001	11.620	<0.001
Walking speed (m/s)	1.09 ± 0.22	1.02 ± 0.16	0.88 ± 0.16	<0.001	0.006	10.516	<0.001
**Saccadic eye movement speed (cm/s)**							
0.5 Hz	34.09 ± 7.54	35.03 ± 7.94	37.24 ± 9.71	0.202	0.309	1.207	0.303
2 Hz	166.25 ± 46.06	159.34 ± 43.70	148.53 ± 36.82	0.167	0.439	1.361	0.260
3 Hz	157.54 ± 37.87	148.23 ± 37.84	133.28 ± 28.25	0.126	0.013	3.736	0.027

MMSE: Mini-Mental State Examination; TUG: timed up and go test; FRT: functional reach test; F distribution was used to determine whether the test is statistically significant by calculating the F value; the *p* value is the probability of obtaining results at least as extreme as the observed results of a statistical hypothesis test.

**Table 2 ijerph-19-07842-t002:** Comparisons of variables between faller and non-faller.

	Functional Reach Test			Timed Up and Go Test			Walking Speed		
	R = 0.186, Adj.R^2^ = 0.011 F = 1.488, *p* = 0.221			R = 0.249, Adj.R^2^ = 0.039 F = 2.723, *p* = 0.047			R = 0.332, Adj.R^2^ = 0.082 F = 4.776, *p* = 0.003		
Dependent Variables	β	t	*p*	β	t	*p*	β	t	*p*
0.5 Hz	0.114	1.287	0.201	0.047	0.537	0.592	−0.047	−0.549	0.584
2 Hz	0.115	1.302	0.195	0.048	0.555	0.580	−0.130	−1.520	0.131
3 Hz	0.083	0.935	0.352	−0.242	−2.772	0.006	−0.282	−3.315	0.001

F distribution was used to determine whether the test is statistically significant by calculating the F value; the *p* value is the probability of obtaining results at least as extreme as the observed results of a statistical hypothesis test.

## Data Availability

The data presented in the study are available on request from the author.

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
