# Peer review of "Decreased Saccadic Eye Movement Speed Correlates with Dynamic Balance in Older Adults"

_ijerph, 2022, doi:10.3390/ijerph19137842_

Round 1

Reviewer 1 Report

In the introduction section, the description of “These findings indicate that SEM may be correlated with and even improve balance ability.” is confusing.

In the methods section, the method of the TUG test is not clearly introduced. By reading this part, I do not konw how man can get the data of TUG during experiment. Besides, the word “patients” is not appropriate, you may mean “participants”. In addition, the unit of 52.8m is certainly not reasonable.

In the results section, Table 1 doesn’t include the data of 3Hz. In addition, the p value of each group of older people is not calculated, which is important for the conclusion that “3 Hz (p = 0.015) significantly decreased more in old older than in young older adults”. Also, some description is not consistent with Table 1. TUG does not decrease when age increases in Table 1. One another mistake is that p value of FRT is 0.092 instead of 0.029 in Table 1.

In the discussion section, the conclusion “TUG, FRT, and WS decreased significantly” can not be obtained from the data in Table 1. (TUG increases with increasing age). Another problem is that there’s too much content of literature in this part, the discussion of this study and experiment itself should be more focused on.

In the conclusion section, I think you mean “induced by” other than “induced for”. Another suggestion is that you can state your conclusion item by item with serial number to make the conclusion clearer and more organized.

Reviewer 2 Report

The authors investigated the quantitative relationship between eye movement and homeostasis in older adults. By designing multiple eye movement experiments and observing the eye movement indicators of a large number of subjects during the experiment, the paper concluded that the response of the elderly to external stimuli will be manifested in the decrease of eye movement speed. To publish the paper, several issues should be extended/explained.

1. The conclusion part is too simplistic. Although the author has done a lot of experimental work, the conclusion part does not summarize the experiments enough.

2. The letters on the title of the table, such as what does F stand for and what does P stand for, are suggested to be explained below the table.

3. The author only compares the differences in eye movements of the elderly between different age groups. We usually think that as a person gets older, his responses in all aspects will gradually slow down. This conclusion is obvious. Haven't other researchers studied this phenomenon? The author should have a comparison or overview.

Reviewer 3 Report

The Materials and Methods section is not completely clear to me. I would appreciate a better organization (Design, Participants, Materials, Procedures, Results analysis).

About point 2.4  Reliability tests, I do not understand what the author did. Are the test already validated? If not, do not use them. If yes, why to investigate the reliability? Besides, this is not a goal of the study. Please, clarify.

In an observational (non-interventional) study, I would not talk about dependent variable. It leads to confusion.  Speed of the target can never be a dependent variable if I understood the methodology and the objectives.

Please, explain better

Thank you

Round 2

Reviewer 1 Report

Thank you for your efforts, there are still some mistakes that should be noticed.

In the Results Part, line 158, “… older age compared with younger age (p = 0.029)”, is the value correct? While in Table 1, p = 0.092.

In the Discussion Part, line 198, “…TUG and WS significantly decreased with age”, you’ve stated that TUG increased as age increased in line179-180.

Author Response

Dear reviewer 1, we appreciated the comments and suggestions, which have helped us to improve the paper. I hope the changes introduced in the manuscript satisfy your requirements.

point 1: In the Results Part, line 158, “… older age compared with younger age (p = 0.029)”, is the value correct? While in Table 1, p = 0.092.

Response: Thank you for your point-out. I apologize for my mistake. 

     I revised to "The FRT was not significant difference."

point 2: In the Discussion Part, line 198, “…TUG and WS significantly decreased with age”, you’ve stated that TUG increased as age increased in line179-180.

Response: Thank you for your point-out. I apologize for my mistake. 

  I revised to "TUG significantly increased and WS significantly decreased with age" 

Thank you for your effort.

Reviewer 2 Report

Inquiries were carefully answered, and the novelty and effectiveness were clarified. Therefore, it was judged to be worth publishing.

Author Response

Dear reviewer

Thank you for your efforts.